# Offering ART refill through community health workers versus clinic-based follow-up after home-based same-day ART initiation in rural Lesotho: The VIBRA cluster-randomized clinical trial

**Alain Amstutz**[1,2,3], **Thabo Ishmael Lejone**[4], **Lefu Khesa**[4], **Mathebe Kopo**[4], **Mpho Kao**[4], **Josephine Muhairwe**[4], **Moniek Bresser**[1,2], **Fabian Räber**[2], **Thomas Klimkait**[2,5], **Manuel Battegay**[2,3], **Tracy Renée Glass**[1,2], **Niklaus Daniel Labhardt**[1,2,3]*

**1** Swiss Tropical and Public Health Institute, Basel, Switzerland, **2** University of Basel, Basel, Switzerland, **3** Department of Infectious Diseases and Hospital Epidemiology, University Hospital Basel, Basel, Switzerland, **4** SolidarMed, Maseru, Lesotho, **5** Molecular Virology, Department of Biomedicine, University of Basel, Basel, Switzerland

* n.labhardt@swisstph.ch

**Data Availability Statement:** A key pseudo-anonymized individual participant dataset collected

## Abstract

### Background

Community-based antiretroviral therapy (ART) dispensing by lay workers is an important differentiated service delivery model in sub-Sahara Africa. However, patients new in care are generally excluded from such models. Home-based same-day ART initiation is becoming widespread practice, but linkage to the clinic is challenging. The pragmatic VIBRA (Village-Based Refill of ART) trial compared ART refill by existing lay village health workers (VHWs) versus clinic-based refill after home-based same-day ART initiation.

### Methods and findings

The VIBRA trial is a cluster-randomized open-label clinical superiority trial conducted in 249 rural villages in the catchment areas of 20 health facilities in 2 districts (Butha-Buthe and Mokhotlong) in Lesotho. In villages (clusters) randomized to the intervention arm, individuals found to be HIV-positive during a door-to-door HIV testing campaign were offered same-day ART initiation with the option of refill by VHWs. The trained VHWs dispensed drugs and scheduled clinic visits for viral load measurement at 6 and 12 months. In villages randomized to the control arm, participants were offered same-day ART initiation with clinic-based ART refill. The primary outcome was 12-month viral suppression. Secondary endpoints included linkage and 12-month engagement in care. Analyses were intention-to-treat. The trial was registered on ClinicalTrials.gov (NCT03630549). From 16 August 2018 until 28 May 2019, 118 individuals from 108 households in 57 clusters in the intervention arm, and 139 individuals from 130 households in 60 clusters in the control arm, were enrolled (150 [58%] female; median age 36 years [interquartile range 30–48]; 200 [78%] newly

during the study, along with a data dictionary, was made available at the time of publication through the data repository Zenodo with open access. DOI link: https://doi.org/10.5281/zenodo.5547573.

**Funding:** This study was predominantly funded by two grants from the Swiss National Science Foundation (IZ07Z0_160876/1 & PCEFP3_181355; https://www.snf.ch/en), both obtained by NDL. AA receives his salary through a grant from the MD-PhD program of the Swiss National Science Foundation (323530_177576). Further funding came through a grant from the Janggen-Pöhn Foundation (http://janggen-poehn.ch/), obtained by AA. The funders had no role in study design, data collection and analysis, decision to publish, or preparation of the manuscript.

**Competing interests:** The authors have declared that no competing interests exist.

**Abbreviations:** ART, antiretroviral therapy; CHW, community health worker; DSD, differentiated service delivery; IQR, interquartile range; SAE, serious adverse event; SMS, short message service; VHW, village health worker; VL, viral load.

diagnosed). In the intervention arm, 48/118 (41%) opted for VHW refill. At 12 months, 46/118 (39%) participants in the intervention arm and 64/139 (46%) in the control arm achieved viral suppression (adjusted risk difference −0.07 [95% CI −0.20 to 0.06]; $p = 0.256$). Arms were similar in linkage (adjusted risk difference 0.03 [−0.10 to 0.16]; $p = 0.630$), but engagement in care was non-significantly lower in the intervention arm (adjusted risk difference −0.12 [−0.23 to 0.003]; $p = 0.058$). Seven and 0 deaths occurred in the intervention and control arm, respectively. Of the intervention participants who did not opt for drug refill from the VHW at enrollment, 41/70 (59%) mentioned trust or conflict issues as the primary reason. Study limitations include a rather small sample size, 9% missing viral load measurements in the primary endpoint window, the low uptake of the VHW refill option in the intervention arm, and substantial migration among the study population.

## Conclusions

The offer of village-based ART refill after same-day initiation led to similar outcomes as clinic-based refill. The intervention did not amplify the effect of home-based same-day ART initiation alone. The findings raise concerns about acceptance and safety of ART delivered by lay health workers after initiation in the community.

## Trial registration

Registered with Clinicaltrials.gov (NCT03630549).

---

### Author summary

#### Why was this study done?

- Community-based antiretroviral therapy (ART) dispensing by community health workers (CHWs) is an important differentiated service delivery (DSD) model in sub-Saharan Africa. However, patients new in care are generally excluded from such DSD models for the first 6 to 12 months.

- Same-day ART initiation during home-based HIV testing campaigns yields improved linkage and engagement in care, but still a third of patients do not link to care within 12 months.

- To date, to our knowledge, involving existing nearby CHWs in drug refills directly after home-based same-day ART start, versus clinic-based refill, has not been evaluated yet.

#### What did the researchers do and find?

- Our open-label, pragmatic cluster-randomized trial in rural Lesotho evaluated ART delivery by an existing lay CHW cadre following home-based same-day ART initiation. In intervention clusters, persons found living with HIV during a door-to-door testing campaign could opt for drug refill by the CHW, with a first routine clinic visit at 6 months.

- At 12 months, 39% and 46% participants in the intervention and control arm, respectively, achieved viral suppression, with no significant difference between arms.

- We found that arms were similar in linkage to care. Engagement in care was non-significantly lower in the intervention arm. Seven and 0 deaths occurred in the intervention and control arms, respectively.

- Of the intervention participants who did not opt for drug refill from the VHW at enrollment, we found that 59% mentioned trust or conflict issues as the primary reason.

## What do these findings mean?

- The offer of village-based ART refill led to similar outcomes as clinic-based refill and did not amplify the effect of home-based same-day ART initiation alone.

- The findings raise concerns about the acceptance and safety of ART delivered by lay health workers after ART initiation in the community.

## Introduction

Of the 38 million people living with HIV, the majority live in eastern and southern Africa [1]. In the last decade, HIV programs in that region have made substantial progress, with 72% of people living with HIV taking antiretroviral therapy (ART) in 2019 [1]. As growing numbers of patients who are taking ART put pressure on already crowded clinics, differentiated service delivery (DSD) models for HIV treatment have been proposed [2]. Such DSD models aim to reduce the frequency of clinic visits, be more client-centered, render services more convenient and less expensive for patients, and thus potentially improve long-term engagement in care [2]. A common HIV DSD model in rural Africa to mitigate the severe shortage of clinical staff is community ART distribution [3]. However, such models are usually reserved for patients established on ART and thus exclude newly initiated patients during their first 6 or 12 months on ART, irrespective of their preferences [3].

Offering home-based same-day ART to individuals with HIV is a promising approach to improve treatment outcomes, but linkage to care remains challenging [4–6]. In November 2018, the WHO held an expert consultation on future ART service delivery priorities and concluded that more research is needed regarding community ART provision and same-day ART initiation [7].

Lesotho, a small land-locked country surrounded by South Africa, has the second-highest adult HIV prevalence globally (22.8%), with more than 70% of the population living in rural areas that are facing a shortage of doctors and nurses [1]. Similar to many countries in the region, a lay health worker network of village health workers (VHWs) has provided community-based primary healthcare services for more than 40 years [8]. We hypothesized that the involvement of VHWs after ART initiation may amplify linkage to and engagement in care by reducing travel costs to the clinic and offering additional psychosocial peer support. So far, to our knowledge, no direct comparison has been conducted of ART refill by an existing community lay health worker cadre following home-based same-day ART initiation versus clinic-based refill.

The pragmatic VIBRA (Village-Based Refill of ART) trial in rural Lesotho evaluated the effectiveness of offering ART refill through VHWs following offer of same-day ART initiation during a home-based HIV testing campaign, compared to standard ART refill at the clinic.

## Methods

### Study design and participants

The VIBRA trial is a cluster-randomized open-label clinical superiority trial conducted in 249 rural villages in the catchment area of 20 health facilities in 2 districts (Butha-Buthe and Mokhotlong) in Lesotho. The 20 health facilities serve a rural population of about 200,000 inhabitants living in a mountainous area with poor infrastructure. Recruitment for the trial lasted from 16 August 2018 until 28 May 2019. A detailed study protocol has been published previously [9].

A home-based HIV testing campaign in the eligible villages, set up for an interlinked trial, called the HOSENG (Home-Based Self-Testing) trial [10], formed the recruitment platform for the VIBRA trial. The HOSENG trial assessed the secondary distribution of oral self-tests to household members absent or refusing to test during home-based testing. Its design and results have been published previously [11]. Prior to the home-based HIV testing campaign, all community councils and village chief councils were visited to get verbal consent to offer the campaign in their villages.

Eligible villages were rural, were confined to the catchment area of the 20 health facilities, had a consenting village chief, and had at least 1 registered VHW who agreed to participate and passed a skill assessment. All community members with a confirmed positive HIV test result (either known HIV-positive or newly tested) and not taking ART were screened by the study nurses for eligibility. Consenting individuals who were 10 years or older, had a body weight of 35 kg or more, had never taken ART (ART-naïve) or had stopped ART more than 30 days prior (ART defaulter), and were physically, mentally, and emotionally able to participate in the study according to the study nurse were eligible. Individuals who planned to get care outside the 2 study districts (e.g., in neighboring South Africa) or were already in care for another chronic disease were excluded.

This trial was approved by the National Health Research and Ethics Committee of the Ministry of Health of Lesotho (ID06-2018) and an ethics committee in Switzerland (Ethikkommission Nordwest- und Zentralschweiz; 2018–00283). It is registered with ClinicalTrials.gov (NCT03630549), and the CONSORT checklist is provided (S1 CONSORT Checklist).

### Cluster sampling and randomization

In collaboration with district authorities, the study team established a list of 648 villages that were eligible for the HOSENG as well as the VIBRA trial. It was not feasible for the study team to visit all 648 eligible villages within the 2 districts. Therefore, an independent statistician created a computer-generated random selection of 159 clusters, proportional to the randomization stratification factors. The targeted sample size of individual participants was not achieved with the initially released clusters; therefore, in November 2019, an additional 144 clusters were randomly selected from the original village list the same way as described above, resulting in a total of 303 villages (clusters) screened for eligibility. After assessing the remaining cluster eligibility criteria, a total of 249 clusters were enrolled in the trial.

Randomization was stratified by district (Butha-Buthe versus Mokhotlong), village size (≥30 versus <30 households), and access to the nearest health facility (easy versus hard to reach, with the latter defined as needing to cross a mountain or river or travel >10 km to reach the health facility). An independent statistician was responsible for the computer-generated

randomization list. This list was uploaded into the study database by the study data manager and provided to the study teams prior to visiting the villages, to enable proper preparation according to group allocation.

We describe the process of cluster selection and randomization in detail in the published study protocols of the HOSENG trial [10] and the VIBRA trial [9].

## Procedures

During the recruitment period, 2 trained campaign teams, each consisting of 6–10 lay counselors, 1 campaign organizer, and 1 supervising study nurse, visited all enrolled villages. In every consenting household, the study team offered blood-based HIV testing and counseling as well as clinical screening for tuberculosis, assessment for harmful alcohol use [12], and HIV prevention services (linkage for voluntary medical male circumcision and condom distribution) to all present household members. Point-of-care blood-based HIV testing followed the national testing algorithm and was offered to all present household members with unknown HIV status. Individuals who tested HIV-positive, or presented proof of known HIV-positive status but were not taking ART, were assessed for eligibility for the VIBRA trial. The study nurse obtained written informed consent in Sesotho, the local language. Illiterate participants provided a thumbprint, and a witness older than 21 years, chosen by the participant, co-signed the consent form. For participants aged <18 years, a literate caregiver older than 21 years provided consent.

Eligible participants in both arms were offered same-day ART initiation at home using an efavirenz-based ART regimen, the national standard first-line regimen at the time of enrollment for a person at least 10 years old and 35 kg. The components of home-based same-day ART initiation are outlined in detail in the published study protocol [9] and consisted of a medical history assessment, a physical examination including WHO staging, 4 point-of-care tests (CD4, hemoglobin, serum creatinine, and cryptococcal antigen [CrAg] test if CD4 < 200 cells/μL), an adherence counseling session, and an ART readiness assessment. Participants with a positive CrAg test or clinical signs suggesting central nervous system involvement were referred to the health facility for ART initiation. If the estimated glomerular filtration rate (eGFR) according to the Cockroft–Gault equation was <50 mL/min/1.73 m$^2$, tenofovir disoproxil fumarate (TDF) was substituted with abacavir or zidovudine, depending on the hemoglobin result. If CD4 was below 350 cells/μL, co-trimoxazole prophylaxis was provided. The study nurse dispensed a 1-month drug supply and scheduled a first follow-up date in 2 weeks.

## Intervention and control

The VIBRA intervention package was designed during a series of workshops with various stakeholders in Lesotho, including community members, clinicians, and district Ministry of Health authorities. These workshops identified community-based ART refill by VHWs after home-based same-day ART initiation as a potentially promising, feasible, and sustainable intervention to improve linkage to and retention in care, for the following reasons. First, the VHWs are rooted and largely respected in their rural community, and refill at village level would minimize travel time and cost to patients. In previous studies conducted in the same setting, VHWs were identified as a trusted cadre for adults and adolescents during community-based HIV testing and counseling [11,13,14]. Further, the 2-year follow-up of participants who did not link to care after being offered home-based same-day ART initiation during the CASCADE trial revealed that a majority did not link because they did not trust, or they disliked, the care provision at their formal healthcare facility [15]. We thus hypothesized that for these persons, follow-up by the VHW could be an attractive alternative. Second, the VHWs

are an already established pillar of the health system and are part of UNAIDS long-term policy, thus offering a sustainable, scalable, and policy-aligned approach [16].

The procedures in each cluster arm are summarized in Fig 1. In villages randomized to the control arm, the study participants were offered the standard of care, i.e., ART refill at the clinic.

In villages randomized to the intervention arm, the participants were offered a 2-component DSD model consisting of ART refill through the VHW and phone text messaging (short message service [SMS]) support as outlined in Fig 1. If ART refill by the VHW was chosen, participants had to attend the clinic only for viral load (VL) measurement at 6 and 12 months after ART initiation. In between, VHWs provided ART refills. At each patient encounter, VHWs followed a paper-based prespecified checklist to assess and document the participants' symptoms regarding potential drug toxicity, opportunistic infection, and immune reconstitution inflammatory syndrome, as well as adherence to ART. If any question on the checklist triggered an alert, the VHW informed the community ART nurse of the corresponding district. The VIBRA trial staff made use of the preexisting monthly VHW meetings at their health facility with a designated facility staff member to exchange reports, address challenges, and coordinate the ART stock refill. Similar to the clinics, the VHWs provided a drug supply for 1–3 months depending on stock availability and participants' preference. The VHWs received a list of all participants who chose their services, and the participants received the VHW's phone number in order to facilitate linkage after initiation.

Participants in intervention villages opting for SMS support were offered a monthly drug adherence SMS reminder and a VL-result-triggered SMS notification after the 6- and 12-month VL measurements. The SMS notifications were sent out automatically through an existing secure platform that is connected to the district laboratory database containing the VL results. Before starting the trial, the VHWs in the intervention clusters completed a 3-day training that covered ART dispensing, clinical symptom screening, adherence assessment, psychosocial support, disclosure, and confidentiality, as well as documentation. Every VHW received a lockable cabinet to store the follow-up checklists of participants and the medications in a safe and confidential way. In addition to their standard monthly stipend from the Ministry of Health (approximately US$15), participating VHWs received monthly calling vouchers and reimbursement of transport costs for attending meetings at the health facility. No additional incentives or fees were paid to the VHWs.

Community mobilization happened through the involvement of all community councils and village chiefs, in collaboration with the responsible VHW. In both arms, tracing of participants lost from care followed the standard procedure at the clinics and was performed by the existing tracing staff at each clinic.

## Data collection

Data were collected and processed in a password-protected electronic database (MACRO, Elsevier). For recruitment and enrollment during the home-based HIV testing campaign, data were entered directly into the database on a tablet. The randomization assignment of the villages was preloaded into the program, and unique household and individual identifiers were automatically generated. For follow-up data during VHW or clinic visits, specific paper-based case-reporting forms served as the source. The study team collected these forms regularly for subsequent data entry. An independent local monitor performed regular source verification checks for all primary and secondary endpoints. The trial additionally underwent independent external monitoring visits by the Ministry of Health of Lesotho and the Clinical Operations Unit of the Swiss Tropical and Public Health Institute. Data closure was on 8 December 2020.

| **VIBRA Control** (Standard of care) | | **VIBRA Intervention** (Offer of VIBRA model) | |
|---|---|---|---|
| 1 | **Offer of home-based same-day ART initiation** | **Offer of home-based same-day ART initiation** | |
| 2 | **Clinic-based ART visit/refill** | **Offer of village-based ART visit/refill** | |
| | 🗓 WHEN — Follow-up interval of 1-3 months | 🗓 WHEN — Follow-up interval of 1-3 months | |
| | 🏠 WHERE — Nurse-led health facility | 🏠 WHERE — At the VHW *Except at 6- and 12-month follow-up: visit at health facility for VL* | |
| | 👤 WHO — Nurse | 👤 WHO — VHW | |
| | 📋 WHAT — Screening for opportunistic infections and ART-related toxicities, adherence assessment, drug dispensing | 📋 WHAT — Screening for opportunistic infections and ART-related toxicities, adherence assessment, drug dispensing | |
| 3 | **No automatic text messages** | **Offer of automatic text messages** | |
| | | 🗓 WHEN — Monthly reminder and after VL measurement | |
| | | 👤 WHO — Automatically sent out from the VL database | |
| | | 📋 WHAT — a) Reminder to adhere to ART b) Message tailored according to VL result | |

**Fig 1. VIBRA trial cluster description.** ART, antiretroviral therapy; VHW, village health worker; VL, viral load.

## Outcomes

The primary endpoint was viral suppression at 12 months, defined as the proportion of all participants in care with a VL below 20 copies/mL at 12 months (range 10–15 months) after enrollment. Over the course of the study, clinics were increasingly collecting blood samples for VL measurement using dried blood spots, instead of plasma, with a lower limit of detection of 400 copies/mL, a level that would be classified as unsuppressed for the prespecified primary endpoint. Therefore, a co-primary endpoint was added during the follow-up period and approved by the ethics committee in Lesotho: viral suppression defined as VL below 400 copies/mL at 12 months (range 10–15 months) after enrollment. Blood draw for VL measurement for all study participants was conducted at the clinics, and the analysis was performed at the corresponding laboratories of the study districts using the COBAS TaqMan HIV-1 Test, v2.0 (Roche Diagnostics).

The secondary endpoints were viral suppression (VL < 20 copies/mL) at 6 months (range 5–8 months after enrollment), viral suppression using the WHO threshold of less than 1,000 copies/mL at 6 and 12 months, linkage to care within 1 month and within 3 months (either with the VHW or at the clinic), engagement in care at 6 and 12 months (clinic visit), all-cause mortality, loss to follow-up for unknown reason, confirmed and unconfirmed transfer of care to a health facility other than the initially attached one at 12 months, and serious adverse events (SAEs). Confirmed transfer out, defined as documented proof of a follow-up visit or laboratory report after transfer, was classified as being engaged in care.

## Statistical analysis

According to a prior home-based same-day ART initiation trial [4], we estimated a recruitment rate of 2–4 individuals per cluster and a proportion of participants engaged in care with documented viral suppression 12 months after same-day ART initiation in the control arm of 50%. We hypothesized that the VIBRA DSD model would increase the proportion with viral suppression by 20% in the intervention arm. Using a conservative intracluster correlation coefficient of 0.015 and power of 80%, a sample size of 262 individuals was needed. The detailed sample size calculation considering different assumptions is in the study protocol [9].

The analysis followed an intention-to-treat approach, including all eligible participants in the clusters irrespective of whether they took up same-day home-based ART initiation and, in intervention clusters, irrespective of whether they opted for the proposed VIBRA DSD model or not. Villages (clusters) were the unit of randomization, whereas individuals were the unit of analysis, with viral suppression as a binary outcome. All participants missing their blood draw or having invalid VL results were classified as not meeting the endpoint of obtaining a valid VL test in the outcome window and being virally suppressed. We analyzed the primary and secondary endpoints with multi-level logistic regression models including village (cluster) as a random effect. We adjusted these models for the prespecified randomization stratification factors (district, size of village, and village access to the nearest health facility) and performed a quadrature check to assess model fit. Results are presented as absolute risk differences with standard errors estimated using the delta method [17]. Differences in the secondary endpoints of mortality, loss to follow-up for unknown reason, transfer out, and SAEs were assessed using Fisher's exact test but not further explored in multi-level logistic regression models due to low numbers of events.

We performed prespecified sensitivity analyses for the primary outcome: using a wider primary endpoint visit window, restricting to participants who attended both the 6-month and 12-month study visit (individual per protocol set), and comparing individuals from the intervention arm who chose VHW ART refill to all participants from the control arm (role of choosing VHW per protocol set). For the primary outcome of viral suppression below 20 copies/mL, we assessed effect modification by prespecified variables and estimated the effect of the HOSENG intervention. All analyses were done using Stata (version 15, StataCorp).

## Results

From 16 August 2018 until 28 May 2019, 124 and 125 villages (clusters) were enrolled in the control and intervention arms, respectively (Fig 2). Two villages from the control arm and 6 from the intervention arm were excluded because their VHWs did not attend the trial-specific 3-day training. In the remaining 241 villages, study teams visited 8,602 consenting households. Among the 14,120 household members encountered during the home-based testing campaign, 2,229 were already known HIV-positive and taking ART, 32 were HIV-positive but not taking ART, 2,126 had proof of testing HIV-negative during the past 1 month or were otherwise ineligible for testing, and 8,868 were eligible for and consented to HIV testing. Among these, 260 (2.9%) tested HIV-positive. Overall, 292 individuals from 130 clusters were found HIV-positive, were not taking ART, and were screened for inclusion in the VIBRA trial. Of the 35 who were ineligible, 25/35 (71%) wished to get care outside the study districts; 4/35 (11%) had a body weight below 35 kg; 4/35 (11%) were physically, mentally, or emotionally not able to participate according to the study nurse; and 3/35 (9%) were already in care for another chronic disease. In total, 257 individuals—118/133 (89%) individuals from 108 households in 57 intervention clusters and 139/159 (87%) individuals from 130 households in 60 control clusters—were enrolled and included in the intention-to-treat analysis.

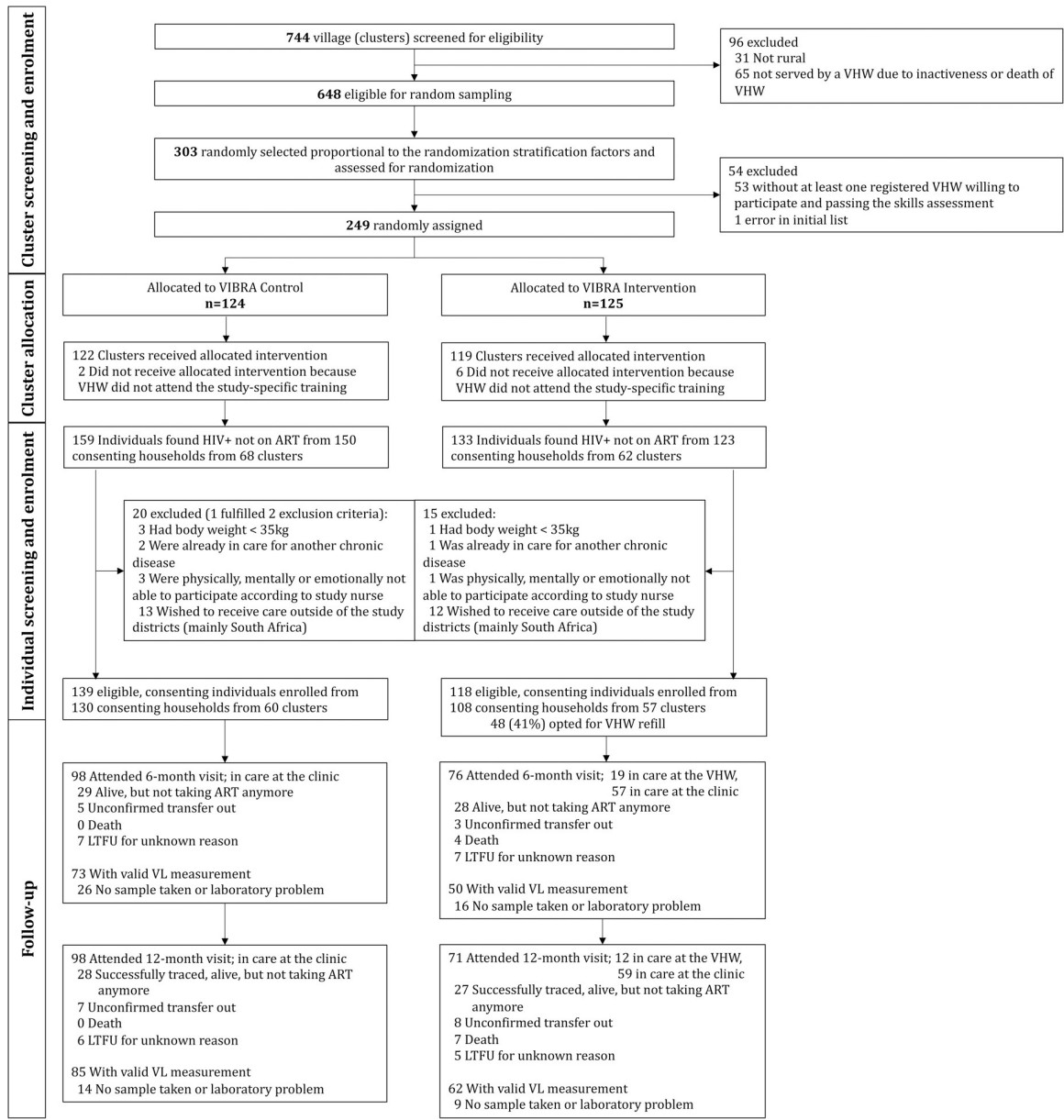

**Fig 2. Participant flowchart.** ART, antiretroviral therapy; LTFU, lost to follow-up; VHW, village health worker; VL, viral load.

The baseline characteristics for the 257 participants are shown in Table 1. The median age was 36 years (interquartile range [IQR] 30–48), and 150 (58%) were female; 63 (25%) had never attended school, and only 39 (15%) reported having employment with a regular income. The majority (183/257; 71%) went to the clinic on foot, with a median 140 minutes round-trip travel time (IQR 80–360). Most participants (99%) were asymptomatic with regard to their HIV infection and had a CD4 cell count of 350 cells/μL or more. Twenty-nine (11%) reported a history of ART exposure (stopped more than 30 days prior to enrollment), 28 (11%) knew of their HIV infection but never took ART, and the remaining 200 (78%) reported a first-time diagnosis. After clinical assessment, the study nurses referred 16/257 (6%) participants to the clinic for ART initiation (10 in the intervention arm and 6 in the control arm) due to clinical,

**Table 1.  Baseline characteristics of trial participants.**

| Characteristic | Control (n = 139) | Intervention (n = 118) | Total (N = 257) |
|---|---|---|---|
| Age, years | 36 (31–47) | 36 (30–49) | 36 (30–48) |
| Sex female | 83 (59.7%) | 67 (56.8%) | 150 (58.4%) |
| Number of children | | | |
| 0 | 22 (15.9%) | 23 (19.5%) | 45 (17.6%) |
| 1 | 22 (15.9%) | 18 (15.3%) | 40 (15.6%) |
| 2 | 29 (21.0%) | 37 (31.4%) | 66 (25.8%) |
| ≥3 | 65 (47.1%) | 40 (33.9%) | 105 (41.0%) |
| Regular sex partner | | | |
| Yes, one | 63 (45.3%) | 69 (58.5%) | 132 (51.4%) |
| Yes, several | 11 (7.9%) | 10 (8.5%) | 21 (8.2%) |
| No | 62 (44.6%) | 37 (31.4%) | 99 (38.5%) |
| Refused to answer | 3 (2.1%) | 2 (1.7%) | 5 (2.0%) |
| HIV status of current partner | | | |
| Don't know | 50 (68.5%) | 63 (79.7%) | 113 (74.3%) |
| Positive and on ART | 16 (21.9%) | 11 (13.9%) | 27 (17.8%) |
| Positive but not on ART | 1 (1.4%) | 1 (1.3%) | 2 (1.3%) |
| Positive but don't know if on ART | 0 (0.0%) | 1 (1.3%) | 1 (0.7%) |
| Recently tested negative | 6 (8.2%) | 3 (3.8%) | 9 (5.9%) |
| Planned disclosure to close person | | | |
| Yes | 134 (96.4%) | 113 (95.8%) | 247 (96.1%) |
| No, never | 1 (0.7%) | 0 (0.0%) | 1 (0.4%) |
| No, not now | 4 (2.9%) | 4 (3.4%) | 8 (3.1%) |
| Refused to answer | 0 (0.0%) | 1 (0.8%) | 1 (0.4%) |
| Education | | | |
| No schooling | 31 (22.3%) | 32 (27.1%) | 63 (24.5%) |
| Primary school | 82 (59.0%) | 61 (51.7%) | 143 (55.6%) |
| Secondary school | 26 (18.7%) | 25 (21.2%) | 51 (19.8%) |
| Years of schooling | 5.0 (2.0–7.0) | 5.0 (0.0–7.0) | 5.0 (1.0–7.0) |
| Employment | | | |
| Employed in Lesotho | 7 (5.0%) | 4 (3.4%) | 11 (4.3%) |
| Employed in South Africa | 5 (3.6%) | 1 (0.8%) | 6 (2.3%) |
| Self-employed with regular income | 18 (12.9%) | 4 (3.4%) | 22 (8.6%) |
| Subsistence farming | 22 (15.8%) | 14 (11.9%) | 36 (14.0%) |
| No regular income | 53 (38.1%) | 64 (54.2%) | 117 (45.5%) |
| Housewife | 32 (23.0%) | 30 (25.4%) | 62 (24.1%) |
| Student | 2 (1.4%) | 1 (0.8%) | 3 (1.2%) |
| Main transportation to health facility | | | |
| Taxi | 22 (15.8%) | 43 (36.4%) | 65 (25.3%) |
| Walk | 110 (79.1%) | 73 (61.9%) | 183 (71.2%) |
| Own car | 3 (2.2%) | 0 (0.0%) | 3 (1.2%) |
| Other | 4 (2.9%) | 2 (1.7%) | 6 (2.3%) |
| Costs of health facility visit | | | |
| Round-trip travel time, minutes | 150 (90–300) | 135 (80–360) | 140 (80–360) |
| Round-trip travel cost, Maloti | 0 (0–20) | 0 (0–40) | 0 (0–34) |
| Any money lost | 89 (64.0%) | 71 (60.2%) | 160 (62.3%) |
| Amount of money lost, Maloti | 35 (25–50) | 40 (30–50) | 40 (25–50) |
| Incur childcare costs | 1 (0.7%) | 6 (5.1%) | 7 (2.7%) |

(*Continued*)

**Table 1.** (Continued)

| Characteristic | Control (*n* = 139) | Intervention (*n* = 118) | Total (*N* = 257) |
|---|---|---|---|
| Alcohol consumption | 46 (33.8%) | 30 (26.8%) | 76 (30.6%) |
| Alcohol abuse[a] | 8 (17.4%) | 8 (26.7%) | 16 (21.1%) |
| Local cannabis use | 25 (18.0%) | 17 (14.4%) | 42 (16.3%) |
| Nicotine smoking | 64 (46.0%) | 56 (47.5%) | 120 (46.7%) |
| HIV/AIDS-related knowledge score | 8 (6–9) | 8 (6–9) | 8 (6–9) |
| HIV/ART history | | | |
| Newly diagnosed | 108 (77.7%) | 92 (78.0%) | 200 (77.8%) |
| Known HIV+/never on ART | 17 (12.2%) | 11 (9.3%) | 28 (10.9%) |
| Previously on ART (stopped >30 days) | 14 (10.1%) | 15 (12.7%) | 29 (11.3%) |
| Prior PMTCT/PEP/PrEP | | | |
| Yes | 2 (1.4%) | 0 (0.0%) | 2 (0.8%) |
| No | 137 (98.6%) | 117 (99.2%) | 254 (98.8%) |
| Don't know | 0 (0.0%) | 1 (0.8%) | 1 (0.4%) |
| Same-day ART prescribed | | | |
| None (referred to clinic for initiation) | 21 (15.1%) | 25 (21.2%) | 46 (17.9%) |
| TDF/3TC/EFV | 102 (73.4%) | 81 (68.6%) | 183 (71.2%) |
| ABC/3TC/EFV | 2 (1.4%) | 4 (3.4%) | 6 (2.3%) |
| AZT/3TC/EFV | 14 (10.1%) | 8 (6.8%) | 22 (8.6%) |
| Reason for referral to clinic | | | |
| Readiness concerns by participant | 15 (10.8%) | 15 (12.7%) | 30 (11.7%) |
| Clinical, lab, or readiness concerns by study nurse | 6 (4.3%) | 10 (8.5%) | 16 (6.2%) |
| How would you remember to take your medication every day? (multiple options possible) | | | |
| Mobile | 102 (73.4%) | 76 (64.4%) | 178 (69.3%) |
| Alarm | 50 (36.0%) | 47 (39.8%) | 97 (37.7%) |
| Person | 80 (57.6%) | 70 (59.3%) | 150 (58.4%) |
| Calendar | 32 (23.0%) | 15 (12.7%) | 47 (18.3%) |
| Timing as other daily activities | 28 (20.1%) | 45 (38.1%) | 73 (28.4%) |
| None | 8 (5.8%) | 9 (7.6%) | 17 (6.6%) |
| Clinical WHO stage | | | |
| 1 | 137 (98.6%) | 116 (99.1%) | 253 (98.8%) |
| 2 | 1 (0.7%) | 0 (0.0%) | 1 (0.4%) |
| 3 | 1 (0.7%) | 1 (0.9%) | 2 (0.8%) |
| History of TB | | | |
| Yes | 5 (3.6%) | 5 (4.3%) | 10 (4.0%) |
| No | 133 (96.4%) | 110 (95.7%) | 243 (96.0%) |
| Current TB treatment | | | |
| Yes | 1 (0.7%) | 0 (0.0%) | 1 (0.4%) |
| No | 137 (98.6%) | 115 (100.0%) | 252 (99.2%) |
| Refused to answer | 1 (0.7%) | 0 (0.0%) | 1 (0.4%) |
| Presumptive TB | 16 (11.5%) | 17 (14.4%) | 33 (12.8%) |
| Cough | 8 (5.8%) | 7 (6.1%) | 15 (5.9%) |
| Weight loss | 13 (9.4%) | 9 (7.8%) | 22 (8.7%) |
| Fever | 7 (5.1%) | 5 (4.3%) | 12 (4.7%) |
| Night sweats | 9 (6.5%) | 8 (7.0%) | 17 (6.7%) |
| On spot sputum collected | 5 (31.3%) | 5 (29.4%) | 10 (30.3%) |
| Other comorbidities | 2 (1.4%) | 0 (0.0%) | 2 (0.8%) |
| Co-trimoxazole prescribed | | | |

(*Continued*)

**Table 1.** (Continued)

| Characteristic | Control ($n = 139$) | Intervention ($n = 118$) | Total ($N = 257$) |
|---|---|---|---|
| Yes | 43 (31.2%) | 27 (23.1%) | 70 (27.5%) |
| No, CD4 ≥ 350 cells/µl | 53 (38.4%) | 47 (40.2%) | 100 (39.2%) |
| No, CD4 not done/results not available | 21 (15.2%) | 18 (15.4%) | 39 (15.3%) |
| No, no ART provided | 21 (15.2%) | 25 (21.4%) | 46 (18.0%) |
| Other concomitant treatment | | | |
| None | 131 (94.2%) | 116 (98.3%) | 247 (96.1%) |
| Traditional herbal medicine | 5 (3.6%) | 2 (1.7%) | 7 (2.7%) |
| Other medicine | 3 (2.2%) | 0 (0.0%) | 3 (1.2%) |
| Weight, kg | 60.0 (54.0–69.0) | 60.0 (53.0–70.5) | 60.0 (54.0–70.0) |
| CD4 count, cells/µL | 365 (250–526) | 411 (254–526) | 386 (253–526) |
| CD4 category, cells/µL | | | |
| <200 | 14 (14.1%) | 19 (22.9%) | 33 (18.1%) |
| 200–349 | 30 (30.3%) | 11 (13.3%) | 41 (22.5%) |
| 350–499 | 28 (28.3%) | 25 (30.1%) | 53 (29.1%) |
| ≥500 | 27 (27.3%) | 28 (33.7%) | 55 (30.2%) |
| Missing | 40 | 35 | 75 |
| Hemoglobin, g/dL | 14.1 (12.6–15.2) | 13.9 (12.5–15.0) | 14.0 (12.5–15.1) |
| Missing | 17 | 15 | 32 |
| Creatinine, µmol/L | 116.0 (78.0–139.0) | 102.0 (72.0–130.0) | 110.0 (75.0–133.0) |
| Missing | 21 | 13 | 34 |
| eGFR[b], mL/min/1.73 m$^2$ | 64.0 (52.0–79.0) | 67.0 (53.0–94.0) | 66.0 (52.0–86.5) |
| Missing | 22 | 15 | 37 |
| CrAg screening | | | |
| Negative | 12 (85.7%) | 18 (94.7%) | 30 (90.9%) |
| Positive | 0 | 0 | 0 |
| Not done | 2 (14.3%) | 1 (5.3%) | 3 (9.1%) |

Results are $n$ (percent of those with non-missing data) for categorical variables and median (IQR) for continuous variables.

[a]Defined as more than 2 positive responses on the CAGE questionnaire.

[b]Estimated using the Cockcroft–Gault equation.

3TC, lamivudine; ABC, abacavir; AZT, zidovudine; CrAg, cryptococcal antigen; EFV, efavirenz; eGFR, estimated glomerular filtration rate; PEP, post-exposure prophylaxis; PMTCT, prevention of mother-to-child transmission; PrEP, pre-exposure prophylaxis; TB, tuberculosis; TDF, tenofovir disoproxil fumarate.

laboratory, or readiness concerns. The remaining 241 participants were offered home-based same-day ART initiation; 211 (88%) were ready to start and thus received a 1-month supply of ART (93/118 in the intervention arm and 118/139 in the control arm). In the intervention arm, 48/118 (41%; 95% CI 32% to 50%) chose the option of ART refill by the VHW, and 78/118 (66%) had confidential access to a cellphone and subscribed to SMS notifications.

At 12 months of follow-up, 110 out of the 257 participants (43%) had a documented VL < 20 copies/mL, 46/118 (39%) in the intervention arm and 64/139 (46%) in the control arm (adjusted risk difference −0.07 [95% CI −0.20 to 0.06]; $p = 0.256$). Using the threshold of 400 copies/mL, 58/118 (49%) and 75/139 (54%) participants in the intervention and control arms, respectively, achieved viral suppression (−0.06 [95% CI −0.18 to 0.07]; $p = 0.369$) (Table 2). These results were consistent across all prespecified sensitivity analyses, including the comparison of individuals who chose VHW ART refill to those in the control arm (S1 Table), and no significant effect modification by any prespecified variable was found (S2 Table).

**Table 2. Primary and secondary endpoints.**

| Endpoint | Total (N = 257) | Control (n = 139) | Intervention (n = 118) | Odds ratio (95% CI)[a,c] | Risk ratio (95% CI)[a,b,c] | Risk difference (95% CI)[a,b] | p-Value[a] |
|---|---|---|---|---|---|---|---|
| **Primary endpoints[c,d]** | | | | | | | |
| VL < 20 copies/mL | 110 (43%) | 64 (46%) | 46 (39%) | 0.73 (0.43 to 1.25) | 0.84 (0.58 to 1.10) | −0.07 (−0.20 to 0.06) | 0.256 |
| VL < 400 copies/mL | 133 (52%) | 75 (54%) | 58 (49%) | 0.79 (0.48 to 1.31) | 0.89 (0.68 to 1.11) | −0.06 (−0.18 to 0.07) | 0.369 |
| **Secondary endpoints[c]** | | | | | | | |
| VL < 20 copies/mL at 6 months | 71 (28%) | 36 (26%) | 35 (30%) | 1.23 (0.70 to 2.16) | 1.16 (0.70 to 1.62) | 0.04 (−0.07 to 0.15) | 0.472 |
| VL < 1,000 copies/mL at 6 months | 112 (44%) | 58 (42%) | 54 (46%) | 1.12 (0.67 to 1.85) | 1.06 (0.76 to 1.36) | 0.03 (−0.10 to 0.15) | 0.665 |
| VL < 1,000 copies/mL at 12 months | 138 (54%) | 78 (56%) | 60 (51%) | 0.77 (0.46 to 1.27) | 0.88 (0.68 to 1.09) | −0.07 (−0.19 to 0.06) | 0.300 |
| Linkage to care | | | | | | | |
| Within 1 month | 133 (52%) | 74 (53%) | 59 (50%) | 0.91 (0.49 to 1.68) | 0.96 (0.70 to 1.21) | −0.02 (−0.16 to 0.12) | 0.757 |
| Within 3 months | 170 (66%) | 90 (65%) | 80 (68%) | 1.16 (0.63 to 2.17) | 1.05 (0.85 to 1.25) | 0.03 (−0.10 to 0.16) | 0.630 |
| Engagement in care | | | | | | | |
| At 6 months | 174 (68%) | 98 (71%) | 76 (64%) | 0.68 (0.39 to 1.16) | 0.88 (0.73 to 1.04) | −0.08 (−0.20 to 0.03) | 0.156 |
| At 12 months | 169 (66%) | 98 (71%) | 71 (60%) | 0.60 (0.35 to 1.02) | 0.84 (0.68 to 0.99) | −0.12 (−0.23 to 0.003) | 0.058 |
| Mortality within 12 months | 7 (3%) | 0 | 7 (6%) | | | | |
| Lost to follow-up at 12 months | 9 (4%) | 5 (4%) | 4 (3%) | | | | |
| Unconfirmed transfer at 12 months | 15 (6%) | 7 (5%) | 8 (7%) | | | | |
| Confirmed transfer at 12 months | 1 (0.4%) | 1 (1%) | 0 | | | | |
| Serious adverse events | 7 (3%) | 0 | 7 (6%) | | | | |

[a]Intervention versus control group, estimated by random effects logistic regression models.

[b]Confidence intervals estimated using the delta method.

[c]Adjusted for stratification factors: district, size of village, and ease of reaching health center.

[d]In total, 23/257 (9%), 14 in the control arm and 9 in the intervention arm, did not have a VL measurement in the primary endpoint window; these were considered not virally suppressed.

CI, confidence interval; VL, viral load.

Table 2 summarizes the secondary endpoint results, and Fig 3 displays the status of care throughout the 12 months of follow-up. Arms were similar in linkage to care, but engagement in care tended to be lower in the intervention arm, although the difference was not statistically significant: At 12 months, 71/118 (60%) were engaged in care in the intervention arm, and 98/139 (71%) in the control arm (−0.12 [95% CI −0.23 to 0.003]; p = 0.058).

Of the overall 88 participants out of care at 12 months, 29 (33%) migrated or transferred out of the study districts, and only 5 (6%) reported structural reasons, i.e., that the clinic was too far away (Table 3). Seven (6%) SAEs, all deaths, occurred in the intervention arm, as compared to 0 (0%) in the control arm. Of the 7 participants who died, 4 had chosen VHW ART refill at enrollment. Out of these, 1 never linked to the VHW or any other care despite several tracing attempts. Two linked to the VHW, clinically deteriorated, and were timely referred to the health facility, but died at the health facility or shortly after discharge. The fourth participant linked to the VHW, was asymptomatic during regular refills at the VHW, and died of a sudden death at home, suggestive of a sudden cardiac event or stroke (S3 Table).

In the intervention arm, 48 and 70 opted for VHW and clinic follow-up, respectively. Fourteen (29%) of the participants who opted for drug refill by the VHW never linked to care (VHW or clinic), whereas 27 of the 70 (39%) who opted for clinic refill never linked to care. At

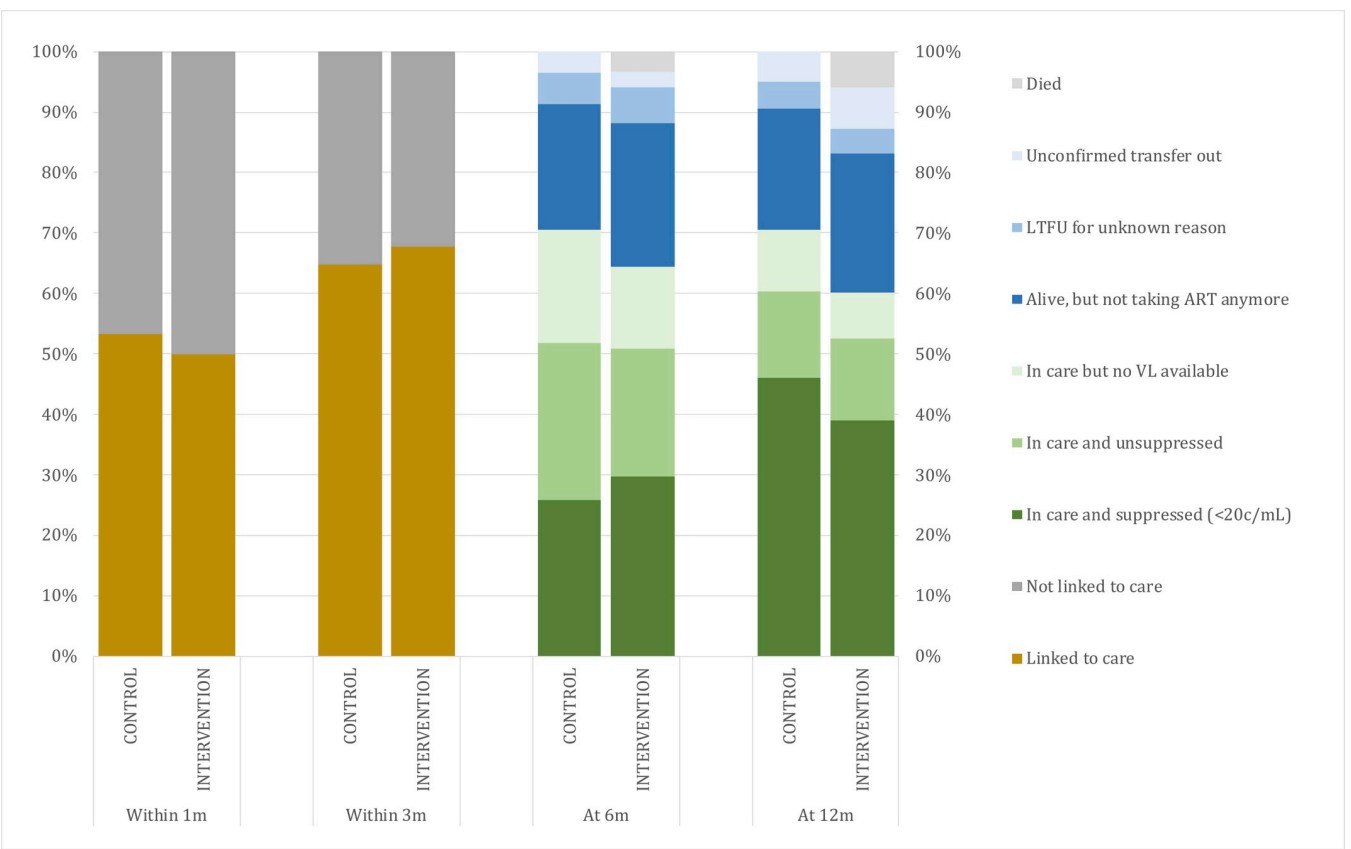

**Fig 3. Status of care at 1, 3, 6, and 12 months of follow-up.** ART, antiretroviral therapy; c/mL, copies per milliliter; LTFU, lost to follow-up; VL, viral load.

12 months of follow-up, among those who had opted for drug refill by the VHW, 12/48 (25%) were still followed by the VHW, 14/48 (44%) had transferred to the clinic, and 15/48 (31%) were not in care. Among those who had opted for clinic refill, 38/70 (54%) were in care at 12 months (S1 Fig).

Comparing the baseline characteristics of intervention participants opting for VHW drug refill with those opting for clinic refill revealed that participants who opted for the clinic refill were younger (median 34 years versus 40 years) and more often also declined the offer of same-day ART initiation (31% versus 6%).

**Table 3. Twelve-month tracing outcomes of participants out of care in both arms.**

| Tracing outcome | Total *n* (%) (*N* = 88) |
|---|---|
| Reported to have transferred to a clinic outside the study districts but no confirmation available | 15 (17) |
| Migrated outside the study districts (mainly South Africa) | 14 (16) |
| Alive, not taking ART, reported that clinic is too far | 5 (6) |
| Alive, not taking ART, reported still not ready for ART or refusing services | 15 (17) |
| Alive, not taking ART, other reasons (not feeling sick, claimed tested negative, other) | 12 (14) |
| Alive, not taking ART, no specific reason available | 9 (10) |
| Died | 7 (8) |
| Lost to follow-up for unknown reason | 11 (13) |

**Table 4. Reasons mentioned by the 70 intervention participants who did not opt for VHW ART refill.**

| Reason | Total *n* (%) (*N* = 70) |
|---|---|
| Does not trust the VHW | 33 (47) |
| Conflict with the VHW family | 8 (11) |
| Clinic is more convenient (i.e., nearby or relative already in care there) | 8 (11) |
| Not ready for any services at the moment | 6 (9) |
| No specific reason available | 15 (21) |

VHW, village health worker.

Of the intervention participants who did not opt for drug refill from the VHW at enrollment, 41/70 (59%) mentioned trust or conflict issues as the primary reason (Table 4).

## Discussion

The VIBRA trial was a pragmatic cluster-randomized clinical trial to assess the effectiveness of a DSD model whereby persons found living with HIV during home-based testing could opt for drug refill by the VHW in their village following same-day ART initiation, with a first routine clinic visit at 6 months. The findings of this study do not confirm our hypothesis that the option of ART refill through the nearby VHW would improve engagement in care and viral suppression. Moreover, we found low overall uptake of the VHW refill option, a tendency towards lower 12-month engagement in care, and a numerically higher mortality in the intervention clusters.

In 2017, UNAIDS launched an initiative to recruit 2 million African community health workers (CHWs) with the aim of contributing to its 90-90-90 strategy of ending AIDS and ensuring sustainable health for all in Africa [16]. Out-of-facility ART dispensing in the community is a common HIV DSD model in sub-Saharan Africa, but generally excludes newly initiated patients, thus triggering a research priority in this field [3,18]. The VIBRA trial aimed to fill this knowledge gap while aligning with the UNAIDS policy initiative, and was based on the fact that the majority of the population in our study districts live in rural areas that are hard to access but that have an established network of VHWs who serve as a trusted cadre for the follow-up of HIV self-testing [11,13]. The CASCADE trial, conducted in Lesotho in 2016, demonstrated that the offer of home-based same-day ART initiation was superior to referral to the clinic for ART initiation, but subsequent ART refills were provided at the clinic and only two-thirds of the intervention participants linked to care [4]. The 2-year follow-up of the intervention participants who did not link to care revealed that a majority did not feel like accessing care at a formal healthcare facility [15]. We hypothesized that the involvement of VHWs in continuing care may further amplify engagement in care after ART initiation in 2 ways. First, it may reduce travel time and cost for the participants to access ART. Second, the VHWs may be an additional psychosocial support after ART initiation. However, among the 88 VIBRA participants out of care at 12 months of follow-up, only 6% mentioned structural reasons. And the psychosocial support was counterbalanced by the fact that 59% of the intervention participants refused the VHW for refill due to mistrust of or conflict with the VHW. The participants who opted for clinic refill were younger (median 34 years) than those opting for VHW refill. While it was important to offer a DSD model with both refill options available to choose from, this finding is important for programs decentralizing HIV care to lay health workers.

VHWs, referred to as CHWs in most settings in sub-Sahara Africa, are elected members of a community, are not healthcare professionals, and perform basic services in their communities. Evidence from meta-analyses and randomized trials in the region demonstrates that

community ART delivery through health workers, community pharmacies, or peer groups has the potential to increase viral suppression rates [3,19–21]. Less is known about ART delivery through existing CHWs. Three systematic reviews pooled viral suppression rates from studies that assessed CHW-assisted ART services compared to nurse-led facility-based care [22–24]. Two reviews concluded that CHW involvement improved suppression rates but only included studies evaluating CHW treatment assistance, not ART delivery [22,24]. The other review included data from studies that evaluated drug delivery through a CHW-similar cadre and reported similar viral suppression rates for drug delivery by trained healthcare workers and CHWs [23]. A randomized trial conducted in Dar es Salaam, Tanzania, evaluated community ART delivery through existing CHWs compared to clinic refills and demonstrated that the intervention was non-inferior in terms of viral suppression [25]. Although, using a similar lay health cadre as in our study, this trial was performed in an urban setting and, more importantly, only included patients established on treatment for at least 6 months.

To our knowledge, there are only 2 randomized clinical trials [20,26] that have assessed community ART delivery among newly initiated patients. In a cluster-randomized non-inferiority trial in Jinja, Uganda [26], trained lay health workers delivered ART after an initial 1-month preparation phase at the clinic following ART initiation. Compared to standard clinic-based refill, the intervention showed similar virological failure rates at 6 months [26] as well as comparable mortality rates at 3 years [27]. Importantly, the trial was based at a well-equipped clinic (the AIDS Support Organisation) [28]; ART initiation happened at the clinic, with a 1-month ART preparation period; and the lay health workers were based at the clinic and visited the participants monthly in their homes by motorbike. The 3-arm Delivery Optimization of Antiretroviral Therapy (DO ART) trial conducted in South Africa and Uganda offered community-based drug delivery after starting ART at the clinic (hybrid arm) as well as after starting ART in the community (community arm), compared to standard clinic ART initiation and refills (clinic arm) [20]. At 12 months, the proportion of patients with viral suppression in the community arm was 74%, which was significantly higher than in the clinic arm. Community-based ART was particularly successful among men. However, the community-based ART refill was delivered through mobile vans with dedicated clinical staff, text message appointment reminders, facilitated rescheduling, follow-up monitoring calls, and potentially more intensive counseling [29]—a model that may have limited scalability.

Task-shifting to CHWs and extending the eligibility for community-based ART delivery to newly initiated patients have growing policy support. However, combining both aspects in the VIBRA DSD model may raise concerns. Even though similar mortality rates have been reported among newly initiated patients followed up in the community and in clinics [26], and the causes of death in our study are not clearly linked to mismanagement by any VHW (S3 Table), the imbalance in mortality between the study groups in our trial suggests that decentralizing HIV care warrants close monitoring.

Our trial had several limitations. First, a rather small sample size and limited numbers reaching the primary endpoint precluded a more conclusive analysis of subgroups, for example, among those who already knew their status or among men. Second, 9% of participants in care at 12 months had a missing VL measurement, and were thus classified as having an unsuppressed VL, which may have underestimated the viral suppression rates. The percentage of missing VL measurements was similar in previous studies in the same study district [4,15] and substantially lower than in a recent trial conducted in neighboring districts [19]. Third, high migration and lockdown regulations during the COVID-19 pandemic may have led to an underestimation of engagement in care. Fourth, due to the design of this cluster-randomized trial and its intervention, the recruiters were aware of the allocation. However, to mitigate recruitment bias, the allocation was concealed to the participants using 2 slightly different

consent forms for control versus intervention. As such, the participating households and individuals were aware of being in a study, but not of being in a trial. Fifth, the low uptake of VHW refill in the intervention arm may have resulted in underestimation of the potential intervention effect. However, in the spirit of DSD it was important to offer the choice of VHW or clinic refill in the intervention group, and to analyze the study using an intention-to-treat approach, to assess the real-life effect if the model were scaled up. The per protocol analysis, including only the participants who opted for VHW refill versus the control arm, did not suggest better outcomes (S1 Table). Nevertheless, more formative piloting work would have been beneficial, and more research to assess the clients' preferences for ART refill in rural Lesotho is needed. Sixth, for logistical reasons, we were not able to collect in-depth data on the reasons for refusing VHW refill at enrollment, for example, the underlying cause of mistrust. Further qualitative research is warranted.

To our knowledge, the VIBRA trial was the first randomized trial evaluating ART refill by an existing lay CHW cadre following home-based same-day ART initiation. The strengths of this trial include the pragmatic design based on existing structures in the health system, an intervention requiring minimal resources for scale-up, and a close-to-real-life intervention. Despite a global policy push, further task-shifting of HIV care to an existing lay health cadre did not amplify the effect of same-day community ART initiation alone, and may raise concerns.

## Supporting information

**S1 CONSORT Checklist.**
(DOCX)

**S1 Fig. Flow of intervention participants.**
(DOCX)

**S1 Table. Sensitivity analyses on primary endpoints.**
(DOCX)

**S2 Table. Primary outcome: Effect modification and subgroup analyses.**
(DOCX)

**S3 Table. Detailed reports of the deaths that occurred.**
(DOCX)

## Acknowledgments

We would like to recognize the hard work and valuable contributions of the study staff in both districts, the tireless support of the SolidarMed team in Lesotho, and the District Health Management Teams in the study districts. We thank the trial monitors, the involved health facilities, village authorities, and VHWs for their advice and dedication to this study. Most importantly, we gratefully acknowledge the study participants.

## Author Contributions

**Conceptualization:** Alain Amstutz, Thabo Ishmael Lejone, Josephine Muhairwe, Moniek Bresser, Tracy Renée Glass, Niklaus Daniel Labhardt.

**Data curation:** Moniek Bresser, Tracy Renée Glass.

**Formal analysis:** Tracy Renée Glass.

**Funding acquisition:** Alain Amstutz, Niklaus Daniel Labhardt.

**Investigation:** Alain Amstutz, Thabo Ishmael Lejone, Lefu Khesa, Mathebe Kopo, Mpho Kao, Josephine Muhairwe, Moniek Bresser, Fabian Räber, Thomas Klimkait, Manuel Battegay, Tracy Renée Glass.

**Methodology:** Alain Amstutz, Thabo Ishmael Lejone, Lefu Khesa, Mathebe Kopo, Mpho Kao, Josephine Muhairwe, Moniek Bresser, Fabian Räber, Thomas Klimkait, Manuel Battegay, Tracy Renée Glass, Niklaus Daniel Labhardt.

**Project administration:** Alain Amstutz, Josephine Muhairwe.

**Resources:** Alain Amstutz, Niklaus Daniel Labhardt.

**Supervision:** Alain Amstutz, Niklaus Daniel Labhardt.

**Validation:** Moniek Bresser, Tracy Renée Glass.

**Visualization:** Alain Amstutz.

**Writing – original draft:** Alain Amstutz.

**Writing – review & editing:** Thabo Ishmael Lejone, Lefu Khesa, Mathebe Kopo, Mpho Kao, Josephine Muhairwe, Moniek Bresser, Fabian Räber, Thomas Klimkait, Manuel Battegay, Tracy Renée Glass, Niklaus Daniel Labhardt.

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
