## [Editor Report · Decision Letter 0]

7 Jul 2021

Dear Dr Labhardt, 

Thank you for submitting your manuscript entitled "Offering ART refill through community health workers versus clinic-based follow-up after home-based same-day ART initiation in rural Lesotho: The VIBRA cluster-randomised clinical trial" for consideration by PLOS Medicine.

Your manuscript has now been evaluated by the PLOS Medicine editorial staff and I am writing to let you know that we would like to send your submission for external assessment.

However, we first need you to complete your submission by providing the metadata that is required for full assessment. To this end, please login to Editorial Manager where you will find the paper in the 'Submissions Needing Revisions' folder on your homepage. Please click 'Revise Submission' from the Action Links and complete all additional questions in the submission questionnaire.

Please re-submit your manuscript within two working days, i.e. by Jul 09 2021 11:59PM.

Once your full submission is complete, your paper will undergo a series of checks in preparation for external assessment. 

Kind regards,

Richard Turner, PhD

rturner@plos.org

---

## [Decision Letter · Decision Letter 1]

29 Jul 2021

Dear Dr. Labhardt,

Thank you very much for submitting your manuscript "Offering ART refill through community health workers versus clinic-based follow-up after home-based same-day ART initiation in rural Lesotho: The VIBRA cluster-randomised clinical trial" (PMEDICINE-D-21-02926R1) for consideration at PLOS Medicine. 

Your paper was discussed with an academic editor with relevant expertise and sent to independent reviewers, including a statistical reviewer. The reviews are appended at the bottom of this email and any accompanying reviewer attachments can be seen via the link below:

[LINK]

In light of these reviews, we will not be able to accept the manuscript for publication in the journal in its current form, but we would like to invite you to submit a revised version that addresses the reviewers' and editors' comments fully. You will recognize that we cannot make a decision about publication until we have seen the revised manuscript and your response, and we expect to seek re-review by one or more of the reviewers. 

We hope to receive your revised manuscript by Aug 19 2021 11:59PM. Please email us (plosmedicine@plos.org) if you have any questions or concerns.

Please let me know if you have any questions, and we look forward to receiving your revised manuscript. 

Sincerely,

Richard Turner, PhD

rturner@plos.org

So as to comply with PLOS' data policy, you will need to arrange a mechanism which does not involve contacting article authors for readers wishing to inquire about data access (https://journals.plos.org/plosmedicine/s/data-availability).

Please add a new final sentence to the "Methods and findings" subsection of your abstract, which should begin "Study limitations include ..." or similar and should quote 2-3 of the study's main limitations. 

In the author summary, please use the active voice (e.g., "We found that ...") in at least one point.

Please rename figure 1 "Participant flowchart" or similar.

Where quoting 95% CI in which one bound is less than zero, we suggest separating the bounds with "to" so that minus signs are easily interpreted.

Please avoid claims of "the first" and the like, and where necessary add "to our knowledge" or similar.

Throughout the text, please adapt reference call-outs to the following style: "... initiation [7].".

In the reference list, italics should be converted into plain text. 

Please list 6 author names rather than 3, followed where appropriate by "et al.".

Please include a completed CONSORT checklist as a supplementary file, labelled "S1_CONSORT_Checklist" or similar and referred to as such in your Methods section. 

In the checklist, please refer to individual items by section (e.g., "Methods") and paragraph number, not by page or line numbers as these generally change in the event of publication. 

Comments from academic editor:

The referees' comments, in different ways, really all relate to the justification for the intervention, the small numbers, the difficulty to generalise to other settings and the use of strict p-value criteria rather than interpreting the results with more insight.

Overall this looks like a major revision, to see if the next resubmission would be more insightful and helpful for the wider HIV programme community. It is the case that sustainably engaging people into HIV/ART care remains a challenge, and using community-based ART is seen as a likely way forward - this makes it all the more important for this paper to put the results in their context, and to discuss how results can (or cannot) be generalised to other settings.

Reviewer 3 questions whether this trial was the correct one to embark on - and I would agree that there should be more detailed justification for the trial design. For example, from the results of the previous trial in the same setting, what lessons were learnt for lack of access/engagement in ART care? In that previous trial, CASCADE, only 2/3 participants in the intervention (home-based ART initiation followed by clinic ART care) engaged in care. What were the reasons in that trial that people did not access ART care, and why did the authors decide that the solution would lie in making ART available in the community from time of initiation onwards (rather than have ART stabilised at the clinic before making refills available in the community). Was a trial with three arms contemplated? So, better justification is definitely required, and then the discussion needs to get into the possible reasons for the small difference between the arms in the current trial.

The ~60% in the intervention who did not take up the offer of community-based ART care are important - who were they? what happened to them, did they access the standard of care instead and how did they adhere then? If they fell out of care completely then the solution to improving ART care needs to be sought elsewhere.

I share Reviewer 2's concerns re regression model used in analysis, and the small numbers involved. The small numbers preclude drawing of reliable conclusions - only 41/188 participants opted for VHW-based refill, with unbalanced drop-out between the arms. Was there selection bias?

Table 2 shows that engagement in care at 12 months (measured by VL assessment) was higher in the routine care arm than in the intervention arm, ,with an OR of 0.6 and a P 0f 0.058, which is borderline significant, and begs the question of whether the intervention would have been detrimental.

The mortality difference between arms is worrying, as is the SAE difference - both worse in the intervention arm (I is not clear to me that the causal relationship in the deaths is adequate - most of it is unknown, and could well have been associated with not being in regular care).

Reviewer 2 asks for a per-protocol analysis, and I agree that that would be of interest. It is currently not clear what happened with those in the intervention arm who declined VHW-based refill. Were they more or less likely to be VL-suppressed at 12 months, were the 7 deaths in the intervention group in this 60%, or in the 40% who accepted VHW-refills?

In sum, there is some work to be done to make this paper of more interest to a wider readership.

Comments from the reviewers:

*** Reviewer #1: 

Statistical review

This paper reports a cluster randomised trial investigating an intervention using village based lay workers to initiate ART amongst individuals who are HIV positive.

The trial appears to have been well-conducted, with the results not showing an advantage of the intervention. I have some minor comments, listed below.

1. Abstract: I'd recommend the secondary outcomes are reported in a similar manner to the primary outcome.

2. Line 174: can the authors say more about the approach used to stratify the randomisation? 

3. Line 250: I'm not sure it is clear in the statistical analysis section how dried blood spot samples that were below limit of detection were treated for the original viral suppression endpoint of <20. Were they treated as non-suppressed (as implied in line 280)?

4. Results/table 3 - it would be useful for future research to report the ICC for the primary endpoint.

5. Results - I would recommend a safety paragraph that reports any adverse events and the 7 deaths that occurred. Were these deaths linked to the intervention at all? 7 in one arm and 0 in the other is, I think, statistically significant. Mortality is listed as a secondary outcome but the authors appear to not analyse it as one.

6. I'd recommend a CONSORT checklist (extension to cluster randomised trials) is provided.

James Wason

*** Reviewer #2: 

Overall

This study describes the result of a cluster randomized trial of an offer of community based ART refill after home based initiation compared to clinic based refill. The question is of importance and the study appears to have been well executed. Still there are some concerns about the methods that would need to be addressed before this was suitable for publication.

Major Points

While the sample size is appropriate from a power standpoint, the cluster randomized design with small(ish) numbers leaves concerns about the comparability of the results. There are differences between populations at baseline (not dramatic, but they exist) and as such, were any of the baseline variables specified a priori as variables that should be controlled in the final model? It does not appear most of what is somewhat imbalanced in Table 1 is adjusted for in Table 2.

The primary endpoint was defined as being virally suppressed an IN CARE. It isn't clear how this was defined since the home based refill arm (for those who chose it) didn't go to the clinic. Did the definition differ by arm?

The most interesting finding in this study is that in the intervention arm, less than half wanted home based refill. The challenge is in figuring out what this generalizes to. The sample size is fairly small and somewhat selected (though not overly so) making it a challenge to determine who this generalizes to. It would help if the authors would discuss this more.

The main results tell us that just under half of the participants had a suppressed viral load at follow up, but we need to know how many had a viral load.

The authors comment that the differences between arms are not significant. This is true but the sample size is small, and the study was powered on 20% differences. Surely a 10% reduction in viral suppression would be seen as a strong negative against home based refills and here we see a 7% difference. So I would not consider these result to be equivalent.

I'm not clear on why no formal comparison (RD, 95% CI) is provided for mortality. The differences are concerning.

Given your outcomes are common, use of logistic regression is exaggerating differences between groups in your relative comparisons. Relative risk models (log binomial) would be more appropriate here.

While the sample size is small, I would still like to see some kind of per protocol analysis that assessed the effect of actually taking up home based refill. This is challenging to do because it was an offer not a requirement and the numbers uptaking the offer reduce the sample size substantially but even if the results end up very imprecise, it would be helpful to know if those who chose home based refills had better results (adjusted appropriately) than those who did not.

Minor points

In the abstract, for consistency, put the order of results the same for each (intervention then control).

We are told that in the intervention arm, 41% opted for home based refill. Can you provide a confidence interval for this finding?

*** Reviewer #3: 

In the study investigators examine the effect of offering home based into retroviral therapy refills after home-based therapy initiation in Lesotho as compared to home-based therapy initiation with facility-based follow up. Overall, 157 adults were randomized and no differences were found in viral suppression after six or 12 months.

While the notion of follow up in the community is important, I think there are several problems with this trial that attenuate its utility. 

First, there is not enough justification provided to motivate the home based follow up. In other words, the choice of an intervention must be well justified for a study to be compelling. I am not saying there is no rationale, but the authors have not provided a clear rationale. Is it based purely on distance? Is there some element of the clinic that is unwelcoming and intimidating? Is this all about the opportunity costs? Can the authors say more about the rationale for follow up at home? In particular given the fact that structural barriers were not commonly reported in the previous work. In other words, if the justification for follow-up at home was due to stigma and or distrust at the healthcare facility then a village health worker-based approach would be in theory useful. But the others have not provided an underlying theoretical reason. 

The need for a theoretical or evidence based justification comes to a head because the majority of patients randomized to community follow up did not opt for village health worker follow up because they felt like they did not trust the village health worker. This represents a problem for the study for several reasons. First, if the hypothesis is about location of services, and the real barrier is the cadre, then the study has tested the wrong intervention. Furthermore, it basically seems that they offered an intervention that was not needed in this setting. But that does not mean the study provides evidence that home based follow up is not useful or effective, which is how this study will be interpreted. 

So, consider a scenario in which patients did in fact trust the community health worker. In the case the intervention of home-based delivery could have worked. So, the fact that there were no effects in this trial is not an indictment of the notion of home-based treatment but rather of the way in which the relationship was created. In other words, the main scientific issue is really the nature of the relationship with the health care worker rather than the location of services. This comes back to the justification for the trial intervention. The absence of a justification makes the fact that people didn't want what was offered 

If this boils down to the fact that people didn't want to follow up in the community with VHW, it seems like that could have been known with adequate formative work and therefore avoided the conduct of an expensive trial. 

The paper is also thin on mediation and how and why trust was low. Other implementation outcomes would be useful - acceptability. Was the issue trust of the VHW in general or trust of any health professional coming to the home? Was it trust about disclosure or trust that the VHW was competent? Do CHW provide other services and do people trust those? Such details would help create some contextual understanding of the results, perhaps using some mixed methods.

***

[LINK]

---

## [Decision Letter · Decision Letter 2]

23 Sep 2021

Dear Dr. Labhardt,

Thank you very much for submitting your revised manuscript "Offering ART refill through community health workers versus clinic-based follow-up after home-based same-day ART initiation in rural Lesotho: The VIBRA cluster-randomised clinical trial" (PMEDICINE-D-21-02926R2) for consideration at PLOS Medicine. 

Your paper was discussed with our academic editor and re-seen by the reviewers. The reviews are appended at the bottom of this email and any accompanying reviewer attachments can be seen via the link below:

[LINK]

In light of these reviews, we will be unable to accept the manuscript for publication in the journal in its current form, but we would like to invite you to submit a further revised version that addresses the reviewers' and editors' comments fully. We will not able to make a decision about publication until we have seen the revised manuscript and your response, and we may seek re-review by one or more of the reviewers. 

We hope to receive your revised manuscript by Oct 14 2021 11:59PM. Please email us (plosmedicine@plos.org) if you have any questions or concerns.

Please let me know if you have any questions, and we look forward to receiving your revised manuscript. 

Sincerely,

Richard Turner, PhD

Senior editor, PLOS Medicine

rturner@plos.org

In the abstract, please add "The trial was registered ..." or similar prior to the registration number. 

Please make that "... in the intervention arm ..." and similar in the abstract.

Does "WVH" need to be corrected late in the abstract?

Noting reference 11 and others, please ensure that all references have full access details. 

Please remove all iterations of "[Internet]" from the reference list.

Comments from the reviewers:

*** Reviewer #1: 

Thank you to the authors for addressing my previous comments well. I have no further issues to raise.

*** Reviewer #2: 

I am happy with all the edits made except the lack of a formal comparison of the deaths. The authors note that this is a small study and use this as justification for not focusing on differences in baseline characteristics between the arms. This is a reasonable response since, even though I would prefer adjustment for some differences (like CD4 count) the random error summar does account for these. But the fact that you did find important differences in mortality is very concerning and needs to include a formal comparison.

*** Reviewer #3: 

I am satisfied with the reviewer responses. My concern that the trial could be much more informative if we knew more about the reasons for mistrust appears to be something that cannot be addressed. I would encourage some more discussion of this issue, in particular a call out to future researchers to take the issue if patient preferences seriously and position themselves to measure such perspectives especially when the intervention depends on them.

***

[LINK]

---

## [Decision Letter · Decision Letter 3]

30 Sep 2021

Dear Dr. Labhardt,

Thank you very much for re-submitting your manuscript "Offering ART refill through community health workers versus clinic-based follow-up after home-based same-day ART initiation in rural Lesotho: The VIBRA cluster-randomised clinical trial" (PMEDICINE-D-21-02926R3) for consideration at PLOS Medicine.

I have discussed the paper with our academic editor and it was also seen again by one reviewer. I am pleased to tell you that, once the remaining editorial and production issues are fully dealt with, we expect to be able to accept the paper for publication in the journal.

[LINK]

Please let me know if you have any questions, and we look forward to receiving the revised manuscript.   

Sincerely,

Richard Turner, PhD

rturner@plos.org

Requests from Editors:

Please finalize the arrangements for data access and detail these in the submission form. 

At line 60 (abstract), please mention the setting, as early in the Methods section (main text). 

At line 70, please avoid beginning the sentence with "48/118" and adapt the sentence to "In the intervention arm, 48/118 ..." or similar.

At line 72, should that be "adjusted risk difference"?

At line 73, please adapt the text to "... engagement in care was non-significantly lower in the intervention arm (adjusted difference ...". Please make a similar amendment at line 103.

At lines 75 and 104, please make that "... intervention and control arms, respectively.".

At lines 84, 111 and 526, please remove "rather".

At line 102, we suggest removing the p value and instead stating "... with no significant difference between arms." or similar.

Please remove the trade mark at line 283, and any other instances throughout the paper. 

We believe that the trial registration number is quoted twice in the Methods section (main text), and once will suffice. 

At line 437, please remove "the" preceding "UNAIDS".

For references 1 & 16, please specify a specific web page and add an accessed date. 

Noting reference 5, please use the journal name abbreviation "PLoS Med." consistently. 

Noting reference 11, please ensure that all references have full access details. 

In other references, please use the journal name abbreviations "PLoS ONE" and "JAMA".

We suggest moving S1 figure to the main body of the paper. 

Comments from Reviewers:

*** Reviewer #2: 

Thank you for the revisions based on my comment. I appreciate the addition of the formal comparison. My personal feeling is that last sentence added is unnecessary as it make a case that this was not the VHW's fault, but this is not the point of the study, the point is whether the overall intervention is safe. We can't conclude that just because your evaluations led you to conclude that the VHW was not responsible doesn't mean that there is not increased risk associated with the intervention. That said, I don't see that what you have said is wrong, so I will leave it to the editors to decide if they want any additional changes based on this point.

***

[LINK]

---

## [Editor Report · Decision Letter 4]

6 Oct 2021

Dear Dr Labhardt, 

On behalf of my colleagues and the Academic Editor, Dr Newell, I am pleased to inform you that we have agreed to publish your manuscript "Offering ART refill through community health workers versus clinic-based follow-up after home-based same-day ART initiation in rural Lesotho: The VIBRA cluster-randomised clinical trial" (PMEDICINE-D-21-02926R4) in PLOS Medicine.

PRESS

Sincerely, 

Richard Turner, PhD 

rturner@plos.org